# A digital dashboard for reporting mental, neurological and substance use disorders in Nairobi, Kenya: Implementing an open source data technology for improving data capture

Daniel M. Mwanga[1,2]*, Stella Waruingi[3], Gergana Manolova[4], Frederick M. Wekesah[1,5], Damazo T. Kadengye[1], Peter O. Otieno[1,6], Mary Bitta[7], Ibrahim Omwom[1], Samuel Iddi[1], Paul Odero[1], Joan W. Kinuthia[1], Tarun Dua[4], Neerja Chowdhary[4], Frank O. Ouma[1], Isaac C. Kipchirchir[2], George O. Muhua[2], Josemir W. Sander[8,9,10,11], Charles R. Newton[6,7], Gershim Asiki[1,12], on behalf of the EPInA Study Team[¶]

1 African Population and Health Research Center (APHRC), Nairobi, Kenya, 2 Department of Mathematics, University of Nairobi, Nairobi, Kenya, 3 Mental Health and Substance Use Program, Nairobi City County Government, Kenya, 4 Department of Mental Health and Substance Use, World Health Organization, 5 Julius Global Health, Julius Center for Health Sciences and Primary Care, University Medical Center Utrecht, Utrecht University, the Netherlands, UMC, AIGHD, the Netherlands, 6 Kenya Medical Research Institute-Wellcome Trust, Kilifi, Kenya, 7 Department of Psychiatry, University of Oxford, Oxford, United Kingdom, 8 Department of Clinical & Experimental Epilepsy, UCL Queen Square Institute of Neurology, London WC1N 3BG, 9 Chalfont Centre for Epilepsy, Chalfont St Peter SL9 0RJ, United Kingdom, 10 Stichting Epilepsie Instellingen Nederland (SEIN), Heemstede 2103 SW, the Netherlands, 11 Neurology Department, West China Hospital, Sichuan University, Chengdu 610041, China, 12 Department of Women's and Children's Health, Karolinska Institute, Stockholm, Sweden

¶ Membership of EPInA Study Group is provided in the Acknowledgements.
* dmwanga@aphrc.org, mtaimwanga@gmail.com

## Abstract

The availability of quality and timely data for routine monitoring of mental, neurological and substance use (MNS) disorders is a challenge, particularly in Africa. We assessed the feasibility of using an open-source data science technology (R Shiny) to improve health data reporting in Nairobi City County, Kenya. Based on a previously used manual tool, in June 2022, we developed a digital online data capture and reporting tool using the open-source Kobo toolbox. Primary mental health care providers (nurses and physicians) working in primary healthcare facilities in Nairobi were trained to use the tool to report cases of MNS disorders diagnosed in their facilities in real-time. The digital tool covered MNS disorders listed in the World Health Organization's (WHO) Mental Health Gap Action Program Intervention Guide (mhGAP-IG). In the digital system, data were disaggregated as new or repeat visits. We linked the data to a live dynamic reproducible dashboard created using R Shiny, summarising the data in tables and figures. Between January and August 2023, 9064 cases of MNS disorders (4454 newly diagnosed, 4591 revisits and 19 referrals) were reported using the digital system compared to 5321 using the manual system in a similar period in 2022. Reporting in the digital system was real-time compared to the manual system, where reports were aggregated and submitted monthly. The system improved data quality by providing timely and complete reports. Open-source applications to report health data is feasible and acceptable to primary health care providers. The technology improved real-time data

**Data Availability Statement:** All the data used in the analysis are within the paper and its Supporting Information files.

**Funding:** This research was commissioned by the National Institute for Health Research (grant number NIHR200134 awarded to CRN) using Official Development Assistance (ODA) funding. The views expressed in this publication are those of the author(s) and not necessarily those of the National Health Service, the National Institute for Health Research or the Department of Health and Social Care. The funders had no role in study design, data collection and analysis, decision to publish, or preparation of the manuscript.

**Competing interests:** The authors have declared that no competing interests exist.

capture, reporting, and monitoring, providing invaluable information on the burden of MNS disorders and which services can be planned and used for advocacy. The fast and efficient system can be scaled up and integrated with national and sub-national health information systems to reduce manual data reporting and decrease the likelihood of errors and inconsistencies.

## Author summary

We present methodological adaptations using open-source data science tools to improve data capture for mental health data in Nairobi. This system was developed against a manual reporting tool Nairobi City County used to capture disaggregated mental health data between 2020 and 2022, which was cumbersome and prone to errors. Leveraging advances in data science and visual analytic technologies, we digitised the reporting tool. We linked to a live dashboard to display the data in a summarised manner for quick decision-making. We present the process we undertook to build the data reporting form and how we integrated it into the live dashboard. We discuss the impact of digitisation on reporting, data quality, and policy decisions in Nairobi City County. We aim to contribute to the advances in data science by demonstrating the utility of open-source platforms to advance data systems and data use for evidence-informed decisions in Africa's digital transformation age.

## Introduction

The World Health Organization (WHO) estimates that over 110 million people in Africa live with mental health disorders [1]. For local planning and prioritisation of mental health care, it is essential to capture service data routinely collected when providing care. The primary reporting system for service data in most low- and middle-income countries (LMICs) is the District Health Information System (DHIS-2) [2] known in Kenya as the Kenya Health Information System (KHIS). The availability of quality and timely data on mental Health from DHIS-2 is a challenge in many African countries[3,4]. In Kenya, the reporting of mental, neurological and substance use (MNS) disorders from KHIS is low, and available data in the KHIS are not disaggregated for the different MNS disorders. Currently, there are only three MNS indicators (generalised as mental disorder, alcohol use and epilepsy) reported in KHIS [5].

The Nairobi City County mental health program developed a reporting tool which prioritised the reporting of MNS disorders guided by the WHO priority MNS disorders in the mhGAP-IG version 2.0 [6]. In addition to the conditions listed in the mhGAP-IG, Nairobi City County also reports data on perinatal mental health disorders (PMHD). This was informed by Nairobi having one of the largest obstetric hospitals with high users flow and thus the need to capture data on PMHD. Secondly, with the roll-out of free maternity services in Kenya in 2013, there has been an increase in utilisation of maternity services and skilled deliveries [7] and Nairobi City County has integrated PMH into the Mother and Child Clinics.

The tool captures demographic profiles of individuals and whether the case was new or a repeat visit at the time of visiting the primary healthcare facility. Reporting new cases is crucial for surveillance purposes and for tracking trends. This is also important for the measurement incidence of a condition. Previously, in each sub-county, the local mental health focal persons reported and aggregated data monthly and submitted to the county mental health focal person

via email. These records were then collated manually to obtain estimates for the County. The above outlined manual reporting process used at the Nairobi City County had challenges including late submissions, inaccurate aggregations, incomplete data and difficulties in analysing the data for timely use for local planning and decision-making. Due to these challenges, it was imperative to devise an efficient method, in the form of a digital tool, that could be used easily by primary healthcare workers to capture and update reports promptly. This was based on the requirement of the county mental health team of a a more efficient way to report mental health data to improve outcomes. This led to the development of the this digitised/online reporting platform.

Thus, this study aims to demonstrate the use of open-source applications for reporting health data using a case study of mental health data reporting in Nairobi. We describe how we digitised the data capture system and linked it to an online live reproducible dashboard built on R Shiny package in R software [8].

## Materials and methods

### Study setting

This study was part of the Epilepsy Pathway Innovation in Africa (EPInA) project in Kenya, Tanzania and Ghana [9]. EPInA mandate is to improve epilepsy treatment pathways, including prevention, diagnosis, treatment and awareness in Africa. Here, we focus on the development of tools to improve data capture for mental health data. It is a case study of mental health program data systems under the Directorate of Health, Nairobi City County, Kenya. It focuses on mental health data reported in all the health facilities under the Nairobi City County. The health facilities managed by the Nairobi City County are primarily public health facilities from level II to level IV. The data covers people of all ages.

### Structure of the Nairobi City County mental health program data systems

The health sector at Nairobi City County has two management levels: The County Health Management Team (CHMT) and the sub-County Management Team (SCHMT), with ten administrative sub-counties. The mental health program is headed by a mental health focal person coordinating mental health services in the County through the SCHMT. Each sub-County is assigned a mental health focal person whose qualifications and practice are in mental health (such as a psychiatrist, mental health nurse, psychiatric physician and psychologist) as per Kenya Mental Health Act (Cap 248) [10]. The mental health focal personnel offer mental health services, coordinate them, and support and supervise healthcare workers at the facility level within their sub-counties. This program is responsible for the management and reporting of MNS disorders in the County, with 54 health facilities seeing people with mental illnesses.

Service data from healthcare facilities were reported monthly through the sub-County focal persons to the county focal persons as aggregated data. These data, under three indicators generalised as mental disorder, alcohol use and epilepsy, were initially reported through KHIS. This required the aggregation of monthly data on paper at the facility level and the entry of aggregated data into KHIS at the sub-County level. Nairobi City County later introduced reporting of aggregated data with more MNS disorders, but it was reported by email, compiled by the SCMH focal person, and sent directly to the county focal person. Simultaneously, aggregated data on the three mental health indicators in KHIS continued to be reported. On average, 18 facilities would report data monthly through the described structures, changing from around 10 in some months to about 20 out of 54 facilities.

### Digitising data capture and reporting platform

We digitised the reporting tool by developing a short electronic data capture form on Kobo Toolbox [11]. This is an online replica of the tool the mental health focal personel used to collect and aggregate data monthly, manually. In the context of this study, manually means aggregating data at the end of the month and submitting using manual forms or on email. A cohort of 100 primary health workers drawn from 54 public health facilities under the Nairobi City County were trained to use the digital tool. The training covered performing data entry, submitting the data to the online server, and troubleshooting in case of system failure. The tool was tested during the training and during the first month of deployment to ensure no system bugs and technology hitches persisted.

The trained healthcare workers currently use digital tools to capture information on all diagnosed cases for real-time data transfer to electronic storage. The form is accessed on an Android tablet or as a link on a web browser. In addition to MNS disorders on the WHO's mhGAP-IG version 2.0, and the added perinatal mental health, the form also collects information on the following items: date when the person was seen at the health facility, sex, age, whether it is a new visit or repeat visit, the disorder diagnosed (multiple disorders are allowed) and the care/service provided (treated or referred).

### Data security and protection

The platform adheres to data security measures in the Kenya Data Protection Act of 2019 [12]. To use the online reporting platform, every primary healthcare worker and the mental health focal worker were given a unique username and password to submit the data only using the front end of the reporting tool (see FigA1 in S1 Appendix). The back-end (with the programmed tools and the database) is only accessed by the programmer and the data manager (see FigA2 in S1 Appendix). The head of the mental health department is the only one responsible for providing authorisation to access the platform.

### Integration between R Shiny dashboard and Kobo Toolbox

Data received in the online database was linked to a live dynamic dashboard built using R Shiny [13] to summarise the data in the form of charts and tables for quicker visualisation and use of the data for decision-making. R Shiny web application (interchangeably called Shiny dashboards) is a visualisation platform that executes in R software [8]. It is a powerful and flexible interactive environment for statistical computing and conducting research in general. The choice of using R Shiny for this study was guided by its reproducibility, open source applicability and flexibility as it can be installed on different platforms, including local servers, cloud servers and personal computers, allowing for easier sustainable access compared to commercial visualisation platforms. Documentation about R Shiny is published elsewhere [13]. Data management processes was conducted in R using various R functions available in its repository of packages designed to handle different data management or analysis needs.

Integration between R Shiny and Kobo toolbox was implemented through an application programming interface (API) using a package called "*robotoolbox*" available in Gitlab or Github mirror [14]. The package has not yet been published in CRAN but is accessed using the remote environment within R. The API is built in Kobo Toolbox and together with the official API instructions, can be accessed from the Kobo account [11]. Once the connection between R and Kobo was established using the API, the data were downloaded into R for management, exploration, and visualisation. The actual codes on how we integrated R and Kobo are provided in S2 Appendix. The requirements for the development of the dashboard include

having R and RStudio installed on any computer operating system. Documentation of the different sections of the dashboard code is documented elsewhere [15].

## Data management, analysis and visualisation

Before displaying summaries on the dashboard, data management processes including formatting and variable labeling are done. These include date formatting using the lubridate package, renaming and labelling variables, recoding variables that need to be recoded, and converting variable types from numeric to string or character where necessary. Analysis involved creating summary tables using functions available in *tidyverse* package [16] and visualisations using either *ggplot()* function found in the *ggplot2* package [17] or using *plot_ly()* function found in the *plotly* package[18]. After the data was cleaned and ready for analysis, the shiny dashboard was created. We used the *shinydashboard* package [15]. It has three parts namely, header, sidebar and body. The dashboard back-end basic structure of the code is given in S2 Appendix. The syntax shinyApp(ui, server) is the function that connects the user interface (UI) and the server sections.

## Hosting of the dashboard

Shiny applications can be hosted in the shinyapp.io platform [19], an easy and self-service platform that enables users and developers to share their R Shiny applications on the web. An alternative approach is to host the dashboard on a local server running on Linux operating system. Open source or no- or low-cost hosting is preferred for sustainability purposes. Currently, the dashboard is hosted on a local server and a backup system on shinyapps.io. Details of how to host the dashboard on shinyapps.io or on a local server are described in S2 Appendix.

## The Nairobi mental health reporting dashboard

The dashboard has four menu items on the sidebar: overall summaries, summary tables, trend analysis and sub-County reports (see FigA3-A5 in S1 Appendix). The menu items enable the user to navigate to the different pages to visualise. At the top of the dashboard is the header and below the header is the body where visuals are displayed. The displays show the overall summaries from all the 10 sub-counties in Nairobi City County. A dropdown menu allows users to select a specific sub-County to visualise separately.

Value boxes are provided to display the overall number of cases submitted and disaggregated by new cases, repeat visits and referrals. This gives the user timely insights regarding the number of new cases reported. This data can be disaggregated by year or month. This is useful for generating quick reports on the reporting for progress and monitoring purposes.

In the second section of the overall summary menu, the reported cases are displayed using bar graphs to show the total number of cases by sub-County, facility where the person was seen, age, sex and type of visit (see FigA3 in S1 Appendix). This helps report caseloads of mental health cases and make essential decisions on allocating resources. A user can download data into a comma-separated (csv) file format for further analysis (see FigA5 in S1 Appendix). The downloaded data are free from any personally identifiable information (PII). The data can be disaggregated regarding facility, type of visit, age, sex, year, month and week. Further disaggregation is possible and the code can be updated to enable this depending on the level of analysis needed.

For monitoring and evaluation purposes, the third menu labelled trend analysis (see FigA5 in S1 Appendix) enables the admin or a monitoring and evaluation staff to monitor the monthly reporting of cases. The data can also be customised by sub-County for planning, monitoring, supervision and learning.

## Pilot testing the digital tool

The system was launched in August 2022 and piloted for three months. During this period, healthcare workers from 54 facilities were trained on how to use the system, and the mental Health focal persons submitted data while continuing to use the manual system. Key lessons learnt during the pilot testing period included improving user-friendliness and ease of access of the data entry interface and submission procedures. Since the tool was accessed mainly online through a web browser, we trained the healthcare workers to create a bookmark for the link on the browser, refresh the link in case of a new form update, and troubleshoot procedures. We also learnt that sending out a notification with instructions on how to refresh the link whenever a new form update is available is necessary. There were visits to the teams at their facilities to provide technical support and document any challenges and lessons.

## Assessment of use of the digital tool

From January to August 2023, the system was adopted as the primary platform for reporting disaggregated mental health data in Nairobi City County. This is in addition to reporting nationally through the KHIS on the three required indicators. All 54 health facilities handling mental health in the County are expected to report. We assessed the use of the digital tool based on the efficiency in reporting, completeness of data, timeliness, user friendliness and how it aligned with the FAIR (Findable, Accessible, Interoperable and Re-usable) principles. For trends, completeness/missing and timelines, we compared the number of cases reported using the manual system in 2022 against those reported through the digital tool in 2023. To assess the difference between the numbers reported between the two systems, we done comparative t-tests of the means of number of facilities reporting per month and the total number of cases reported every month. While the difference in numbers may be contributed to some extent by natural temporal trends, they indicate the effect of the digitisation of the data system. This is triangulated with information from qualitative case narratives collected, as described below.

Qualitative information was also obtained from focal persons using the system through case narratives from key informants who mainly included focal persons representing health care workers in the sub-counties. We also had a data review meeting in December 2022 where we compiled best practices, lessons and challenges experienced while implementing the system. The main thematic areas were their experiences using the system and their perceptions of how the system has improved different aspects of data collection, monitoring and use for decision-making. Data were analysed thematically, and quotes summarising feedback on this were extracted and reported verbatim in the paper.

## Ethical considerations

The study was part of the EPInA project approved by the Scientific Ethics Review Unit (SERU) of the Kenya Medical Research Institute (KEMRI) (Reference Number: KEMRI/RES/7/3/1). The data used to demonstrate the effect of the digitization are from routine service data from the Nairobi City County health facilities submitted by sub-County mental health focal persons as part of their routine health care services. As we used anonymised data and did not collect new data, the study did not require institutional review board approval. For the case narratives from the focal persons who were the direct users of the dashboard, verbal consent was obtained.

## Results

### Results from the pilot phase

Between August and December 2021, 3381 individuals were diagnosed with at least one mental health illness and manually reported from 20 facilities, with a mean of 15 facilities per month ranging from 10–20. During a similar period in 2022, 4956 diagnosed with at least one mental health disorder (2343 new, 2399 repeat visits and 14 referrals) were reported after the digital system was launched, mean of 29 facilities per month ranging from 23–32 (p<0.001) (Table 1). August -December 2022 was considered a testing phase for the system, and the data reported may not be complete. This was when the focal persons and the mental healthcare providers were familiarizing themselves with the system until the data review meeting held in December 2022, where lessons were shared based on the experiences from the pilot phase.

The main lessons learned included the need for conducting regular support visits to the facilities, sharing best practices among the users, and prompting feedback in the event of questions asked by the team. The feedback was shared by the team members having a question directly calling their supervisors. They compiled their questions and shared them with the developer or shared directly through shared social media group explicitly created for this work.

There were no significant challenges relating to access to the dashboard. However, based on the experiences after implementing the system for three months, the team reported an increased workload specifically as they had to report each individual seen in real-time. To improve the system's user-friendliness, they recommended additional features in the dashboard, including customised sub-County reports and additional indicators at the community level.

### Results from the dashboard compared to manual system during full implementation phase

This is a live dashboard where data are continuously uploaded. To compare the digital and the manual system, we restrict analysis results to data reported between January to August 2023 (digital system) and between January to August 2022 (manual system). This is shown in Table 2.

**Table 1. Comparison of all cases reported between August and December 2022 in the digital system and between August and December 2021 using the manual system during the *pilot phase*.**

| Month | Digital System (August-December 2022) | | | | | | Manual System (August-December 2021) | | |
| --- | --- | --- | --- | --- | --- | --- | --- | --- | --- |
| | Number of facilities (n = 54) | Cases per facility | New cases§ | Repeat visits§ | Referral§ | Total ξ | Number of facilities (n = 54) | Cases per facility | Total |
| August | 30 | 35 | 499 | 540 | 5 | **1044** | 20 | 43 | **864** |
| September | 29 | 32 | 491 | 449 | 3 | **943** | 19 | 48 | **903** |
| October | 30 | 43 | 692 | 575 | 6 | **1273** | 14 | 47 | **661** |
| November | 32 | 33 | 541 | 511 | 0 | **1052** | 10 | 51 | **508** |
| December | 23 | 28 | 320 | 324 | 0 | **644** | 10 | 45 | **445** |
| **Total** | **40** | **123 ¥** | **2543** | **2399** | **14** | **4956** | **20** | **169¥** | **3381** |
| Mean | 29 | 34 | 509 | 480 | 3 | **991** | 15 | 47 | **676** |
| Minimum | 23 | 28 | 320 | 324 | 0 | **644** | 10 | 43 | **445** |
| Maximum | 32 | 43 | 692 | 575 | 6 | **1273** | 20 | 51 | **903** |

Statistical tests: difference is mean of facilities reporting per month (p<0.001), cases per facility per month (p = 0.002) and total number of cases reported (p = 0.051);

Notes: § denominator is the total number reported in the period denoted by ξ; and ¥ means total number of cases reported by a facility on average during the period

**Table 2. Comparison of all cases reported between January and August 2023 in the digital system and between January and August 2022 using the manual system during *full implementation phase* among 54 health facilities.**

| Month | Digital System (January-August 2023) | | | | | | Manual System (January-August 2022) | | |
|---|---|---|---|---|---|---|---|---|---|
| | Number of facilities (n = 54) | Cases per facility | New cases§ | Repeat visits§ | Referral § | Total ξ | Number of facilities (n = 54) | Cases per facility | Total |
| January | 22 | 45 | 484 | 505 | 5 | **994** | 14 | 45 | **626** |
| February | 27 | 50 | 742 | 607 | 5 | **1354** | 16 | 44 | **707** |
| March | 24 | 55 | 672 | 637 | 3 | **1312** | 12 | 44 | **530** |
| April | 24 | 39 | 434 | 507 | 0 | **941** | 18 | 44 | **786** |
| May | 20 | 60 | 587 | 605 | 0 | **1192** | 20 | 43 | **864** |
| June | 19 | 49 | 427 | 501 | 2 | **930** | 15 | 38 | **574** |
| July | 18 | 70 | 625 | 637 | 2 | **1264** | 19 | 40 | **752** |
| August | 17 | 63 | 483 | 592 | 2 | **1077** | 12 | 40 | **482** |
| **Total** | **38** | **239 ¥** | **4454** | **4591** | **19** | **9064** | **22** | **242 ¥** | **5321** |
| Mean | 21 | 55 | 567 | 584 | 2 | **1153** | 16 | 42 | **671** |
| Minimum | 18 | 39 | 427 | 501 | 0 | **930** | 12 | 38 | **482** |
| Maximum | 27 | 70 | 742 | 637 | 2 | **1354** | 20 | 45 | **864** |

Statistical tests: difference is mean of facilities reporting per month (p = 0.004), cases per facility per month (p = 0.007) and total number of cases reported (p<0.001);

Notes: § denominator is the total number reported in the period denoted by ξ; and ¥ means total number of cases reported by a facility on average during the period

Thirty-eight of 54 facilities (71%) reported mental health data through the digital platform between January and August 2023 compared to 22/54 (41%) facilities that reported data from January to August 2022 through the manual system. The mean of facilities reporting per month was 21, ranging from 17–27 for the digital system 16 ranging from 12–20 using the manual system (p = 0.004). The number of facilities reporting was not constant across the months for the digital and manual systems, with some months having only a third (32%) of the facilities reporting in the digital system and a quarter (23%) in the manual system. Nine-thousand-sixty-four people diagnosed with at least one mental health disorder (4454 new, 4591 repeat visits and 19 referrals) were reported through the system compared 5321 through the manual system between January to August 2022 (p = 0.007). The number reported through the digital system is almost double of those reported using manual system. This is mainly due to the increased number of facilities using the digital system. The manual system did not distinguish between the number of new vs repeat visits.

In addition to the actual numbers reported through the two systems, we show in Table 3 the utility of the digital system on different aspects of data processing and utilisation compared to the manual system.

## Views of focal persons

Two representatives of the focal persons were also interviewed on their experiences using the system and whether it is helpful. Their narratives showed that the digital approach has improved the timeliness of reporting, supervision and data use for clinical decision-making. Below are excerpts of what they said verbatim;

*"Since August 2022 the reporting of mental health indicators in Nairobi City County has been made easier. APHRC team developed a digital Reporting system where a mental health practitioner is able to upload data immediately after they see a patient. In the sub-County where I work we have a new facility offering alcohol rehabilitation services and also other mental health services on demand. Incorporating this new facility on the digital reporting system has*

**Table 3. A comparison between the digital system and manual system on different aspects of data use.**

|  | Digital system | Manual system |
|---|---|---|
| ***Data quality*** |  |  |
| Timeliness | Data entry and submission was done real-time or within two days on average after diagnosing and treating an individual | Data entry was done on paper, entered into Microsoft Excel and submitted at the end of the month to the county mental health focal person within the first week of the month on average |
| Completeness | 38 facilities reported data between January and August 2023 through the digital system. | 29 facilities reported data between January and August 2022 through the manual system |
| Mining data for decision making | Summarised data are displayed in the form of tables, charts, and value boxes in real-time, enabling quick decisions by the mental health management team at the county level. Through the system, the county team have used the data to advocate for additional resources including staffing and have used it for routine updates in county meetings. | Data mining involves summarising individual facility reports submitted via emails, which is a time-consuming, often a tedious process that can be prone to mistakes. This makes analysis and use of data for decision making cumbersome. |
| Support supervision | The monitoring and evaluation team are able to view progress of reporting and determine facilities lagging behind. They can also customise reports for each facility or sub-County as they plan their support supervision visits. | Analysing data to determine which facilities or sub-counties need support was tedious and time consuming as outlined in the mining process above. This reduces the ability to make timely informed decisions on which areas need more support. |
| ***FAIR Principles***[20] |  |  |
| Findability | Data are stored in an online database with controlled access. The data represent each individual. | Data are available in fragmented manner since each individual facility or sub-County sends through email. To find the data, one has to compile the different facility reports. The data are available in aggregated format. |
| Accessibility | Access to the data is controlled and must be authorised by the County's head of mental health. | Access to the data is controlled and must be authorised by the County's head of mental health. |
| Interoperability | The system can be integrated with other systems using application programming interface upon authorisation by the head of mental health | Integration is not easy. |
| Re-Usability | Data are more readily available for use compared to manual system. This is routine data on mental health collected to inform planning and policy decisions on mental health in the County. The data can be used for different purposes, including research. | Availability of the data for use or re-use might take longer since one has to compile the different facility reports sent on email. |

*made it easy for the new staff to upload data daily and carry out data review meetings to identify areas of focus during community sensitisation on mental health and demand creation for services. Through supportive supervision of the mhGAP trained physicians more mental health cases are being reported which informs decision on employment of more mental health practitioners by the county government."* Focal person 1, Nairobi City County

Another sub-County focal person also had similar sentiments but also added that the data are now easier to interpret including being able to interpret trends of the different MNS disorders reported through the system. The focal person had the following to say;

*"The dashboard has played a crucial role in making informed decisions on treatment strategies to adopt for the various mental illnesses and also presents the data in a way that is easy to interpret showing the trends and patterns of the various categories of mental illness and also helps visualise activities across the various sub counties providing mental health services and also to monitor the staff actively reporting and follow up on the staff who are not reporting".* Focal person 2, Nairobi City County

The system also helped reduce workload and made data entry work more efficient, making it possible for healthcare workers to see more people. Two of the focal persons had this to say;

*"Daily reporting helps in preventing backlog of data which can be cumbersome at end of the month. Integration of mental Health in primary health care facilities has seen more patients*

*seeking services and this data reflects on the digital system as some non-specialist health care workers have had their capacity built and have rights to report digitally."* Focal person 1, Nairobi City County

*"Before the acquisition of dashboard Nairobi mental health data was being collected haphazardly as the available MOH reporting tools only capture two variables in relation to mental health i.e. psychiatry illness, alcohol use disorder and epilepsy. This disadvantaged many practitioners who were not able to clearly capture the various categories of mental illnesses they were attending to in the facilities which led to massive underreporting of mental illnesses and thus portrayed an in accurate picture. With the advent of the dashboard the staff are able to report on the specific mental illnesses and this has led to a clearer picture of the illness categories and helped staff to direct more efforts in specific areas of interest in the various sub counties as the dashboard is able to provide a near real time picture of the actual situation in terms of data."* Focal person 2, Nairobi City County

While the system has been shown to have improved several aspects of mental health data use in the County, it is still being continuously updated and improved from time to time. After one year of implementation, the system will undergo several adjustments and refinements drawn from the lessons learnt so far. These include additional features to allow more interactivity and better support supervision, generating individual sub-County customised reports and additional indicators to track community activities, such as counselling sessions.

## Discussion

We showed the use of open-source applications for reporting health data using a case study of mental health data in Nairobi. Overall, there was an increase in the number of facilities reporting data on mental health, from less than half using manual system between January to August 2022 to over two-thirds using digital systems between January and August 2023. The number of mental health cases reported through the digital system between January and August 2023 was almost double that reported using the manual system in the same period in 2022. The digital system also allowed data disaggregation between new and repeat cases and referrals, while the digital system reported all as an aggregate. Data was reported through the digital system at the individual level, while the data submitted through the manual system was at the aggregate level. Analysis of feedback from users of the system, represented mainly by the sub-County focal persons, showed that the digital system improved the timeliness of reporting, efficiency and use of data for timely decisions. The data from the system have been used for advocacy and for providing current information during routine department meetings.

The number of health facilities reporting was not constant over time, with variations observed over time. The first few months after launch, the numbers were high and but slightly reduced with time. During the pilot phase (August to December 2022), we actively visited the teams, checking about areas needing improvementand what support was needed. This could explain the decline in the number of facilities seen from January 2023, when we reduced the number of visits and check-ins with the team. The decline could also be attributed to the fact that the system was still new and users were still familiarizing themselves with it, since changes takes time. This indicates the need for continuously conducting support supervision and checking-in with the teams to identify areas where support is needed.

We have shown that it is feasible to use open-source, low-cost electronic data capture platforms integrated with data science technologies like R Shiny to improve decision-making data. With advances in data science tools, data analysis, and visualisation technology, simple

localised solutions like ours can help improve data availability. Our study used R Shiny [13], a package in R software [8], to build the visualisation. Shiny applications have been used in multiple disciplines. Funk and Zahadat (2021) used it for reproducible dissemination of public health data [21]. For their study, they developed an R Shiny dashboard that visualised cancer data from George Washington University. Their approach to building an interactive dashboard is similar to ours. Still, ours uses an application programming interface (API) to link the dashboard with live data from the health facilities submitted to an online database.

R shiny has also been used for planning healthcare capacity, comparing health metrics geographically and, mining health records data [22], and comparing health inequalities for different countries by the WHO [22]. R Shiny has also been used to provide a platform for power and sample size calculations for longitudinal models in R [23]. Our study adds to this development by demonstrating the ability of R Shiny integrated with Kobo Toolbox electronic data capture tool to improve data capture of health data. We did not find another study that has implemented R Shiny combined with Kobo toolbox. Our study shows that R Shiny can be used to visualise routine data submitted to an online database for quick real-time decision-making. The fact that the code is reproducible makes it easier to scale up such interventions to other counties or replicate them for reporting data in other disciplines.

The application can be hosted in the shinyapps.io platform for simple routine processes. However, for sensitive and complex datasets, it is recommended that the application be hosted in local servers secured with solid firewalls. The process for hosting the dashboard in a local server is outlined in this paper. This can be a more cost-effective hosting method if the system requires the development of more than five applications. The hosting infrastructure provided by Posit (shinyapps.io) is limited to a maximum of five applications[19].

The real-time dashboard data makes it possible to identify caseloads and particular mental health indicators trends by facility and sub-County. It enables County and sub-County mental health teams to make informed decisions about appropriate intervention, resource allocation and staffing needs. This is an important aspect of how simple systems and solutions can greatly contribute to evidence generation and informed decision-making. The cost of implementing open-source applications is only the personnel and computing costs (hardware and storage) since the software itself is free. The personnel costs may include costs to pay staff time and training the users of the system. This means that it can be used in any setting where software subscription is not a guaranteed option. While more sophisticated platforms can be built, our study demonstrates that simple open-source solutions can be used in resource-poor settings.

The KHIS has made significant efforts to create more mental health indicators. The process is ongoing and will take longer before it is fully implemented. To prevent missing out on reporting disaggregated mental health data during the period, our system provides an alternative to provide needed mental health data that could otherwise not have been reported efficiently. Our system offers individual-level data, and reporting is real-time. Our system also demonstrates that simple open-source solutions are an excellent option to consider in local contexts where data are needed rapidly. It has shown that it is an efficient alternative where data on specific indicators is required but may not have been captured in national data systems. The next steps in the development of the system include continuous maintenance and incorporating feedback from the users to its improvement including addition of spatial analysis. The spatial analysis component will provide spatial distribution of the data based on geographic information system (GIS) approach, which would be useful for targeting resource allocation. The county mental health team is also providing feedback to the teams updating the KHIS platform based on the lessons learned from this platform.

This system can be replicated elsewhere, and the methodology can be adapted for design in other software such as Python. We developed our system using the open source Kobo

Toolbox [11] for data capture and integrated it with R Shiny package in R [16] using application programming interface (API) for the visualization and reporting. To replicate this system, one can can use the guidelines provided in this paper, alongside available support online through community platforms such as stack overflow for developing R scripts for data management [15] and R Shiny[16], available in the R software documentation pages online. The data capture can be built using any other electronic data capture platforms as long as they have APIs to enable integration.

## Strengths and limitations

This study has several strengths. First, it utilises open-source platforms, which are low-cost. This provides a cost-effective way of reporting routine data in income-restricted settings. Second, it is sustainable because the primary healthcare workers reporting through the system are employees of the County, which means they continue reporting as part of their responsibilities. Another strength of this system is because it is scalable as descrived above.

The study, however, has limitations. This study was conducted during COVID-19 pandemic. The country did not have a formal mental health response plan including aspects of data capture [24], and in addition, infection with SARS-CoV-2 exacerbated mental health illnesses across the age spectrum [25] as the pandemic evolved. This study, however, does not provide data on mental health cases that may be due to the effects of COVID-19. Challenges in reporting may be varied, including system-related and human resources issues. Our study focuses on the former by proposing methods that can be used to navigate the challenges of reporting health data from public health facilities.

Developing the dashboard requires advanced coding skills in R. This can be a limitation if such skills are lacking within an institution. We have documented the steps in building the dashboard for capacity building the county teams. The dashboard and data reporting require internet connectivity for the data to synchronise. It might be a challenge for places where internet connection is not guaranteed. This can be addressed by deploying offline forms on tablets to collect data offline and synchronising when internet connectivity is available. Recent announcements by Kobo suggested a change in the terms of service [11]. Kobo announced that the platform will continue to be free (open-source) for over 90% of its current users. Still, some circumstances will attract subscription fees, for example, when one exceeds 5000 monthly submissions. While this can be a limitation for small-scale use in the future, it will still be an alternative for capturing data for rapid response or as a temporary solution to fill in a gap or technical hitches in the national systems. When mental health records are digitised in all health facilities through the national system, the lessons learnt in this solution can be replicated in developing other digital platforms in other disciplines.

## Conclusion

Building open-source applications to capture and report on MNS disorders is feasible. The technology improved real-time mental health data capture, reporting, and monitoring, made work more efficient by reducing data backlog and enhanced use for quick clinical decision making. Real-time reporting has enabled better support supervision, which will improve data quality. The next step is to scale up this system and integrate the disaggregated reporting of mental health data in the national and sub-national health management information systems. It is recommended that institutions should invest in building the capacity of data managers and producers in resource-restricted settings to build and use open-source platforms for better data capture and reporting. For scale-up and sustainability, health management teams need to be involved in the application development phases to ensure the system responds to the needs

of the health system. Lessons from localised innovations like this should be used when developing or updating more extensive national or subnational data systems.

## EPInA Study Group

Abankwah Junior, Albert Akpalu, Arjune Sen, Bruno Mmbando, Charles R. Newton, Cynthia Sottie, Dan Bhwana, Daniel Mtai Mwanga, Damazo T. Kadengye, Daniel Nana Yaw, David McDaid, Dorcas Muli, Emmanuel Darkwa, Frederick Murunga Wekesah, Gershim Asiki, Gergana Manolova, Guillaume Pages, Helen Cross, Henrika Kimambo, Isolide S. Massawe, Josemir W Sander, Mary Bitta, Mercy Atieno, Neerja Chowdhary, Patrick Adjei, Peter O. Otieno, Ryan Wagner, Richard Walker, Sabina Asiamah, Samuel Iddi, Simone Grassi, Sloan Mahone, Sonia Vallentin, Stella Waruingi, Symon Kariuki, Tarun Dua, Thomas Kwasa, Timothy Denison, Tony Godi, Vivian Mushi, William Matuja.

## Supporting information

**S1 Appendix. Screenshots of different sections of the dashboard.**
(DOCX)

**S2 Appendix. Integration of R and Kobo tool box.**
(DOCX)

**S1 Dataset. Testing phase.**
(CSV)

**S2 Dataset. Implementation phase.**
(CSV)

## Acknowledgments

We recognise the training facilitators, including the Nairobi City County mental health focal person and the sub-County mental health focal persons who went through the training of trainers (TOT) and facilitated the actual mhGAP training for the primary healthcare workers. The contributions of the all mental health focal persons coordinating mental health activities and reporting in the sub-counties is appreciated. Sub-County mental health focal persons included Keziah Anunda, Peter Ichangai, Regina Muthoni, Paul Mbugua, Judith Anyango, Nuru Hussein, Carol John, Hellen Wekesa, Lucy Mwaura and Jecinta Gakuu. The authors also thank the leadership of the Nairobi City County Health Services for supporting the project.

## Author Contributions

**Conceptualization:** Daniel M. Mwanga, Stella Waruingi, Gergana Manolova, Frederick M. Wekesah, Charles R. Newton, Gershim Asiki.

**Data curation:** Daniel M. Mwanga, Damazo T. Kadengye, Ibrahim Omwom, Frank O. Ouma.

**Formal analysis:** Daniel M. Mwanga, Damazo T. Kadengye, Ibrahim Omwom.

**Funding acquisition:** Damazo T. Kadengye, Tarun Dua, Josemir W. Sander, Charles R. Newton, Gershim Asiki.

**Investigation:** Daniel M. Mwanga, Stella Waruingi, Gergana Manolova, Frederick M. Wekesah, Charles R. Newton, Gershim Asiki.

**Methodology:** Daniel M. Mwanga, Stella Waruingi, Gergana Manolova, Frederick M. Wekesah, Damazo T. Kadengye, Peter O. Otieno, Mary Bitta, Ibrahim Omwom, Samuel Iddi,

Paul Odero, Tarun Dua, Neerja Chowdhary, Isaac C. Kipchirchir, George O. Muhua, Josemir W. Sander, Charles R. Newton, Gershim Asiki.

**Project administration:** Frederick M. Wekesah, Joan W. Kinuthia.

**Software:** Daniel M. Mwanga, Paul Odero.

**Supervision:** Frederick M. Wekesah, Damazo T. Kadengye, Samuel Iddi, Tarun Dua, Neerja Chowdhary, Isaac C. Kipchirchir, George O. Muhua, Josemir W. Sander, Charles R. Newton, Gershim Asiki.

**Validation:** Stella Waruingi, Gergana Manolova, Frederick M. Wekesah, Damazo T. Kadengye, Peter O. Otieno, Mary Bitta, Samuel Iddi, Tarun Dua, Neerja Chowdhary, Josemir W. Sander.

**Visualization:** Daniel M. Mwanga, Frank O. Ouma.

**Writing – original draft:** Daniel M. Mwanga.

**Writing – review & editing:** Stella Waruingi, Gergana Manolova, Frederick M. Wekesah, Damazo T. Kadengye, Peter O. Otieno, Mary Bitta, Ibrahim Omwom, Samuel Iddi, Paul Odero, Joan W. Kinuthia, Tarun Dua, Neerja Chowdhary, Frank O. Ouma, Isaac C. Kipchirchir, George O. Muhua, Josemir W. Sander, Charles R. Newton, Gershim Asiki.

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
