## [Decision Letter · Decision Letter 0]

11 Jun 2024

PDIG-D-23-00453

A digital dashboard for reporting mental, neurological and substance use disorders in Nairobi, Kenya: implementing an open source data technology for improving data capture

PLOS Digital Health

Dear Dr. Mwanga,

Thank you for submitting your manuscript to PLOS Digital Health. After careful consideration, we feel that it has merit but does not fully meet PLOS Digital Health's publication criteria as it currently stands. Therefore, we invite you to submit a revised version of the manuscript that addresses the points raised during the review process.

Please submit your revised manuscript within 30 days Jul 11 2024 11:59PM. If you will need more time than this to complete your revisions, please reply to this message or contact the journal office at digitalhealth@plos.org. Please include the following items when submitting your revised manuscript:

We look forward to receiving your revised manuscript.

Kind regards,

Ryan S McGinnis

Academic Editor

PLOS Digital Health

Journal Requirements:

Additional Editor Comments (if provided):

Reviewers' comments:

Reviewer's Responses to Questions

**Comments to the Author**

1. Does this manuscript meet PLOS Digital Health’s publication criteria? Is the manuscript technically sound, and do the data support the conclusions? The manuscript must describe methodologically and ethically rigorous research with conclusions that are appropriately drawn based on the data presented.

Reviewer #1: Yes

Reviewer #2: Yes

2. Has the statistical analysis been performed appropriately and rigorously?

Reviewer #1: N/A

Reviewer #2: Yes

3. Have the authors made all data underlying the findings in their manuscript fully available (please refer to the Data Availability Statement at the start of the manuscript PDF file)?

Reviewer #1: Yes

Reviewer #2: Yes

4. Is the manuscript presented in an intelligible fashion and written in standard English?

Reviewer #1: Yes

Reviewer #2: Yes

5. Review Comments to the Author

Reviewer #1: Summary:

goal is to use shiny/kobo dashboard to view and submit health care data about MNS disorders in real time. the system shows better real-time performance, as it was not aggregated and reported retrospectively in month increments (real time is more timely, accurate, and comprehensive). This work is aimed to build upon a previous manual tool, and to evaluate improvements using this new system

positives:

- i think this is a very relevant work for this venue. 

- this work shows the importance of real-time reporting, data visualization for public health, and how digitization can improve growing health systems

- the authors did well at "setting the scene" with the previous monthly reporting method, describing the challenges with routing data through "focal persons"

-authors did well to describe how the tool was made and why they selected the technologies they did (especially for shiny)

- I appreciated that the dashboard was described verbally in layout (lines 192-198) this improves accessibility/reproducibility

- Pilot testing was well described with lessons learned clearly indicated.

-flowed well and was easy to follow step by step

- table 3 is great and clear to see differences in the systems. 

- i appreciated the qualitative reviews from focal persons, i would be curious to hear if there were any focal persons who did not like the digital system more than the manual system.

changes requested:

-would recommend rework of keywords to be more general

- are there age limits for reporting? is this adults and children? More information on the populations/facilities would be helpful.

- I was surprised to see that many facilities still did not report, even with this new system. It also looked like less and less facilities were reporting as time progressed... can the authors reference this trend or describe why we are still seeing low compliance rates?

-table 1 format is strange and a bit difficult to read without a column divider between the two primary sections. please reformat so we can clearly compare the manual versus digital system.

-spelling error in media on line 269

- i am unclear on what is different in the data being shown in table 1 and table 2 - please make it clearer what we are comparing across these two tables and what makes them different. I can see the data is different but it is not abundantly clear why.

- it would be helpful to understand the financial burden of implementing this new system compared to the previous system (how expensive would this be for another group to do in terms of money, personal, resources, etc). this was touched on a bit in the discussion but could be dived into more.

- i would be curious to hear next steps and what other data is planned to be incorporated next and why

- i feel it is a significant downfall of the work that they don't have a screenshot or visualization of this dashboard or of some of the data collected. please add this in your next revision to help readers understand the dashboard beyond just the verbal description provided within text. 

- can the authors provide any additional numerical/quantitative analyses or statistical tests to show/demonstrate, beyond just summary counts, the significance in additional data being collected with this new system.

Reviewer #2: This paper describes a digital tool for reporting mental health services provided in Nairobi, Kenya. Overall, its well-structured, well-written, and relevant to PLOS Digital Health. I have a few minor issues with the paper that the authors should be able to address easily.

1. Introduction line 86: Why the sudden focus on perinatal mental health disorders specifically? Is the digital health tool being developed specific to PMH disorders (if so, were there any specific considerations made for this specific subset of mental health disorders), or is that the focus of the existing reporting tool? Of note, as far as I can tell, the acronym PMH is not used anywhere else in the paper outside of this paragraph. The authors should motivate the introduction of this concept.

2. Introduction line 96: This sentence is the motivation for this entire work and needs to be backed up in some way. Are there other studies, national/international surveys, government reports, etc. that show that the existing system suffers from “late submissions, inaccurate aggregations, incomplete data and difficulties in analysing the data for timely use…”?

3. Materials and Methods line 132: The authors note that they “digitized the data collection tool” here. Is this the same as the “reporting tool” mentioned and described on line 84? If so, the authors should pick one of the terms and use that terminology throughout the paper. If not, it would be helpful to readers to (at least briefly) describe here what that tool entails so they have some context as to what exactly is being digitized.

4. Materials and Methods line 195: The authors mention that the spatial distribution page is still being developed, what is the spatial distribution page? It isn’t mentioned anywhere else and I’m not clear as to what functionality it is supposed to provide. It doesn’t seem to be providing any functionality key to the evaluation of the system, so perhaps this should be moved to the discussion where the authors describe functionality to be added?

5. Tables 1 and 2: These tables are a bit difficult to parse, and it took me some time to understand what I was supposed to be taking away from the table. I think adding some additional border lines and bolding would go a long way towards guiding readers towards what columns to compare and key takeaways. I also don’t understand first symbol in the notes in these tables. The authors state that this symbol indicates the total number reported in the period for the digital system, does that mean the referrals and total are not the total number reported in the period? For example, my understanding is that the first data row of Table 1 indicates that there were 499 new cases, 540 repeat visits, and 5 referrals during the month of August; does the 5 mean something different?

6. Somewhere in the paper (perhaps in the introduction or discussion?) it would be beneficial to add a sentence or two discussing why having data on new cases, repeat visits, and referrals rather than just the total is useful/valuable to have.

7. Discussion: In thinking of what a reader would take away from this work, I would like to see an additional paragraph/section in the discussion section that clearly lays out the design considerations that were learned by the authors from this work that could be used by readers to develop their own similar system. If design considerations were specific to this system, I’d like some thought/insight into how they might generalize or be used for other systems.

Grammar/Minor Things:

Line 82: Missing period, should be “disorders. Currently, there are…”

Line 92: Missing a subject, who reported and aggregated data monthly?

Line 288: “in the digital system and a quarter (23%).” Should this end with “and a quarter (23%) in the manual system.”?

Table 3: Why is “Month” in the top left cell of this table?

6. PLOS authors have the option to publish the peer review history of their article (what does this mean?). If published, this will include your full peer review and any attached files.

**Do you want your identity to be public for this peer review?** For information about this choice, including consent withdrawal, please see our Privacy Policy.

Reviewer #1: No

Reviewer #2: Yes: Josh Cherian

---

## [Decision Letter · Decision Letter 1]

18 Sep 2024

A digital dashboard for reporting mental, neurological and substance use disorders in Nairobi, Kenya: implementing an open source data technology for improving data capture

PDIG-D-23-00453R1

Dear Mr Mwanga,

We are pleased to inform you that your manuscript 'A digital dashboard for reporting mental, neurological and substance use disorders in Nairobi, Kenya: implementing an open source data technology for improving data capture' has been provisionally accepted for publication in PLOS Digital Health.

Best regards,

Ryan S McGinnis

Academic Editor

PLOS Digital Health

Reviewer Comments (if any, and for reference):

Reviewer's Responses to Questions

**Comments to the Author**

1. If the authors have adequately addressed your comments raised in a previous round of review and you feel that this manuscript is now acceptable for publication, you may indicate that here to bypass the “Comments to the Author” section, enter your conflict of interest statement in the “Confidential to Editor” section, and submit your "Accept" recommendation.

Reviewer #2: All comments have been addressed

2. Does this manuscript meet PLOS Digital Health’s publication criteria? Is the manuscript technically sound, and do the data support the conclusions? The manuscript must describe methodologically and ethically rigorous research with conclusions that are appropriately drawn based on the data presented.

Reviewer #2: Yes

3. Has the statistical analysis been performed appropriately and rigorously?

Reviewer #2: Yes

4. Have the authors made all data underlying the findings in their manuscript fully available (please refer to the Data Availability Statement at the start of the manuscript PDF file)?

Reviewer #2: Yes

5. Is the manuscript presented in an intelligible fashion and written in standard English?

Reviewer #2: Yes

6. Review Comments to the Author

Reviewer #2: I am satisfied with the changes the authors have made to the manuscript. I've just noted some minor grammar issues I noticed while reviewing the revised manuscript.

Line 104: outined should be outlined.

Lines 94/95: Are PMH and PMHD the same? If so I would change one so that the acronym is the same, otherwise the authors should define PMHD so it’s clear that it’s something different.

Line 110: “developing of the this” should be “development of this”

7. PLOS authors have the option to publish the peer review history of their article (what does this mean?). If published, this will include your full peer review and any attached files.

**Do you want your identity to be public for this peer review?** For information about this choice, including consent withdrawal, please see our Privacy Policy.

Reviewer #2: No
